# Mapping National Environmental Sustainability Distribution by Ecological Footprint: The Case of Italy

**Silvio Franco, Barbara Pancino * and Angelo Martella**

Department of Economics, Engineering, Society and Business Organizations, University of Tuscia, 01100 Viterbo, Italy; franco@unitus.it (S.F.); a.martella@unitus.it (A.M.)
* Correspondence: bpancino@unitus.it

**Abstract:** The paper proposes a possible way of spatially representing sustainability in Italy. For this purpose, the ecological footprint approach was used as a methodological framework to assess the level of sustainability of the 8092 Italian municipalities. For each municipality, the exploitation of ecosystem services, assessed by the ecological footprint indicator, and the corresponding availability of biological capacity, associated to an indicator, have been calculated and compared, thus generating a map representing the relative sustainability of Italian municipalities. The results show a very scattered distribution of ecological balance, wherein unsustainable conditions characterize more than 60% of the territory and almost 95% of the Italian population. Despite the limitations of the methodology and some assumptions regarding the ecological footprint assessment at the municipality level, the study represents an attempt to produce an innovating tool that, based on an operational definition of sustainability, can represent natural resource exploitation at the local level, and provide useful information to address coherent and targeted environmental policies of sustainability.

**Keywords:** ecological footprint; Italian municipalities; sustainability map





## 1. Introduction

The concept of sustainability has increased in prevalence in the research sector as well as in public opinion. This situation has induced the widespread use of terms "sustainability" and "sustainable" in many different contexts, but, at the same time, it has generated vagueness about the real meaning of the concept [1,2]. However, from a scientific and economic perspective, the definition of sustainability should be clarified, and become an attribute that can characterize a specific object as sustainable.

To verify whether sustainability features are present in an object, certain conditions must exist [3]: a clear definition of sustainability; one (or more) indicator(s) to operationally apply the definition; the possibility of correctly assessing this (these) indicator(s) in a specific object (for instance, a product, a process, a firm or a region).

Regarding the first aspect, without entering the wide debate on the dimensions (environmental, economic, and social) of the concept of sustainability, their nature and their reciprocal interactions, in this work, we will focus on the relationship between the supply of natural resources and their demand from anthropogenic activities within a territory. In other words, a definition that looks at the environmental dimension of sustainability, intended as the preservation of natural capital, from an economic perspective will be adopted.

With reference to the assessment of sustainability, an indicator based on the ecological footprint approach will be used; a methodology that, as will be argued, is consistent with the definition of sustainability adopted in the study.

Our analysis focused on Italy, a country that, according to its current ecological footprint (4.41 gha per capita) [4], which is some five times higher than its resource-generating capacity (0.88 gha per capita), is strongly unsustainable. At the same time, the country is also characterized by very different local situations in terms of environmental

pressures and natural resources availability. The "structural" unsustainability of the Italian economic system is evident from the trend in its ecological balance time series, which has been negative since the first evaluation in 1961 [4]. This is confirmed in a recent study [5], which reports that Italy "has the fourth-highest per capita EF in the EU 27 countries", and its footprint is mainly created by transportation and food consumption.

In this situation, it is interesting to assess how this unsustainability is spread throughout the national territory by adopting an innovative approach aiming at estimating the ecological balance at the municipality level in a synthetic way. Such information could be useful for constructing national and local policies capable of integrating economic development with the awareness of environmental issues, also through the use of Next Generation funds.

Moving from these considerations, the objective of this study is to build a map of the local sustainability in the Italian territory, assuming as the spatial reference a single municipality. The first section is devoted to a discussion of the assumed definition of sustainability and the theoretical approach, which can assess the existence of such conditions in a territory. In the second section, the methodology adopted for the analysis is presented. Then, the study's outcome, represented by the constructed sustainability map of Italian municipalities, is presented and discussed. The paper ends with some final remarks about the limitations of the study and its possible implications and future developments.

## 2. Background

For the development of our analysis, it was necessary to stick to a definition of environmental sustainability that is operational and can be assessed for a specific "object"; that is, in our case study, the Italian municipalities. The choice of this territorial scale enables us to provide an evaluation that considers as much detail as possible from an administrative point of view. Indeed, the administrative division of the Italian territory consists of 20 regions, 107 provinces and 8092 municipalities, the latter of which represents the most restricted level of administrative bodies.

### 2.1. Sustainability Definition

The concept of environmental sustainability considers natural capital—defined as the set of functions provided by the environment [6,7]—via two different approaches: possible substitution with man-made capital, and strict preservation. These two positions establish the difference between the paradigms of weak and strong sustainability [8].

For neoclassical economists, sustainability is a condition wherein the capital (in a broad sense) is maintained at a constant level [9]; to this end, natural capital can be substituted with man-made capital [10,11]. When the income of an economic activity is reinvested in manufactured or human capital, and its value is greater than the value of the natural capital lost in such an activity, a (weak) sustainability condition is established [9]. As Dietz and Neumayer [8] pointed out, the weak sustainability paradigm represents an extension of the neoclassical approach to economic growth, wherein natural resources were explicitly considered as a factor of production. Specifically, "the Hartwick–Solow models of the 1970s imputed non-renewable and renewable natural resources into a Cobb–Douglas production function, which is characterised by a constant and unitary elasticity of substitution between factors of production. This entailed the assumption that natural capital was similar to produced capital and could easily be substituted for it", [8] (p. 618). Consequently, from a weak sustainability perspective, there are no fundamental differences between the nature of the kinds of well-being that natural and man-made capital can generate [12].

On the other hand, the strong sustainability paradigm is based on the idea that natural capital accomplishes many functions, some of which are not replaceable [6,13]. The functions of natural capital associated with production and consumption processes, such as raw material provision and waste assimilation, can be partially substituted by man-made capital. The same happens for some amenity services, which represent another function of natural capital [8].

However, the basic life support function cannot be substituted [14]. This implies that "the global environmental and ecological system that provides us with the basic functions of food, water, breathable air and a stable climate should be subject to a strong sustainability rule", [8] (p. 619). In addition, some other reasons for natural capital non-substitutability must be considered—the consequences of its depletion are largely unknown and uncertain, and its loss is often irreversible [15,16].

In the strong sustainability approach, the possibility of replacing natural capital with man-made capital is not excluded. However, this option cannot be applied when the level of natural capital exploitation leads to the irreversible destruction of such capital [12]. This is true for those elements of natural capital that make an essential contribution to human well-being [6]. The need to preserve the consistency of these "critical" components of natural capital requires the adoption of a strong sustainability perspective in economics [17].

### 2.2. Sustainability Assessment Using Ecological Footprint

The definition of strong sustainability demands that natural capital be preserved in physical terms. It implies that every empirical analysis intending to evaluate a condition of strong sustainability must be based on a measurement of the physical dimension of the natural capital, in terms of its availability and possible exploitation by economic activities.

Regarding indicators able to perform such a measurement, the literature is extremely wide and diverse; for a discussion on this topic see, among others, [18–22]. In such reviews, many sustainability indicators are discussed, compared, and evaluated; one of the most important is ecological footprint [23]. Indeed, among scholars there is general agreement that the ecological footprint is an indicator that enables a strong sustainability measurement [7].

Consistent with Daly's two principles of strong sustainability [24], the ecological footprint methodology accounts for the demand and supply of the basic resources and ecosystem services that a community needs to support its lifestyle [25]. Monfreda et al. [26] state that the ecological footprint approach follows the core requirements of strong sustainability; Knaus et al. [27] claims that it reflects the principles of strong sustainability; Mori and Christodoulou [21] affirm that it is based on strong sustainability and Huang [28] asserts that ecological footprint is a strong sustainability indicator.

The ecological footprint approach accounts for the level of sustainability of a territory by first assessing the indicator ecological footprint (EF), which expresses the bioproductive area required by the local population to produce the renewable resources and ecological services it uses. This value is then compared with biocapacity (BC), which tracks the supply of renewable resources and ecological services provided by the local ecosystems [25,29].

Such comparison leads to the assessment of an indicator, ecological balance (EB), able to translate in quantitative terms the environmental surplus/deficit situation of a region, and hence to verify its strong sustainability condition. EB is calculated as the difference between the availability of resources available in a region, measured by BC, and the resources consumed by the activities of a local population, measured by EF. If EB is higher than zero, i.e., EF is lower than BC, the carrying capacity of the region is not exceeded, and the region is judged to be sustainable, under a strong sustainability approach [30].

Some authors raise extensive criticisms about both the ecological footprint approach in general, and its full suitability as an indicator of strong sustainability (see for example [30–33] for a discussion of critical and supporting points of view). However, this methodology has been used in different studies aiming at evaluating strong sustainability at the local level; for example, in Germany [27], Australia [34,35], Italy [36], Canada [37,38], the Mediterranean area [39], China [28] and Portugal [40]. All these studies, independently of their quantitative results, show how it is possible to assess the environmental impact of economic activities on a local scale using the ecological footprint.

It is worth considering that in studies evaluating sustainability at a local scale using the ecological footprint method, a different approach to the interpretation of BC can be adopted.

Indeed, the regional EF value can be compared with the average global biocapacity instead of the BC of the region itself [41]. This way of evaluating local sustainability refers to the idea that natural resource functions, in particular the absorption of $CO_2$ emissions, cannot be confined to local ecosystems.

In our case, a direct comparison between local BC and EF seems more coherent with the aim of the study, and consequently this approach will be applied to evaluate the strong sustainability of Italian municipalities.

## 3. Materials and Methods

The assessment, performed for all the (j = 8092) Italian municipalities, is based on the evaluation of the ecological balance (EB) per capita, obtained as the difference between biocapacity (BC) per capita and Ecological Footprint (EF) per capita:

$$EB_j = BC_j - EF_j \tag{1}$$

Although these concepts are now consolidated in the scientific literature, as regards the method of calculation of EF and BC and their meaning, please refer to [23] and to all the subsequent bibliography.

### 3.1. Calculation of $EF_j$

The estimation of the EF per capita at the municipal level ($EF_j$) was carried out considering the most updated value of Italian per capita ecological footprint ($EF_N$), which refers to the year 2017 [4].

Moving from this figure, the ecological footprint of the residents in each municipality was estimated by considering their relative level of consumption with respect to the national one. The assumption of a direct relationship between the level of consumption and the ecological footprint in a region, which is based on the idea that the quantity of purchased goods is strictly related to the exploitation of bioproductive resources demanded by their production, is supported by some studies [42,43].

The local level of consumption is affected by different drivers; in this study, two of them were explicitly considered: (i) the average income of municipality inhabitants and (ii) the regional consumer price index.

Likewise, a similar effect of the price index is quite evident, which has a significant impact on the possibilities of residents' purchases; this is particularly true in a country such as Italy, where economic differences between different areas (namely, the north and south) are quite noticeable. Other variables influencing the ecological footprint, such as the residents' purchasing power or the preferences in consumer expenditures for different products, were not considered in the calculation. Indeed, besides the difficulty of getting a reliable estimation of their value at the municipality level, the first one is strictly linked to the local price index and the second one has a limited influence in determining the aggregate level of consumption.

For each municipality, the index ($I\_Inc_j$) defined as the ratio between the local and the national per capita income [44] was assessed.

Regarding the local price index, even if no data were available from official sources, it was possible to refer to a database created in a recent study [45]. As such data are calculated at the provincial level, the price index applied in our study ($I\_Pr_j$) assumes the same value for all the municipalities within a province.

Then, the per capita ecological footprint in each municipality ($EF_j$) is estimated as follows:

$$EF_j = EF_N \times I\_Inc_j \times I\_Pr_j \tag{2}$$

### 3.2. Calculation of $BC_j$

Following the standard ecological footprint methodology [23], for each municipality (j), the land area $S_{ij}$ that falls into each of the following i = 5 land-use classes was assessed:

1.  Built-up land;
2.  Cropland;
3.  Grazing land;
4.  Forest land;
5.  Water.

By means of GIS software, the calculation was performed by overlapping the municipal borders network with the CORINE Land Cover (CLC) map.

The CORINE (Coordination of Information on the Environment) land cover database reports data on land use in European countries at a high spatial resolution. The CLC project started in 1985, and was coordinated by the European Environmental Agency to produce consistent and reproducible data concerning the state of the environment in the European Community [46]. The CLC databases for years 1990, 2000, 2006, 2012 and 2018 are available. The data are available on a spatial scale of 1:100,000 with a minimum mapping unit of 25 hectares for areal phenomena, and a minimum width of 100 m for linear phenomena [47].

Land cover is organized into 44 classes structured at three hierarchical levels [48]. To assess the area in each one of the five classes for the biocapacity evaluation, the second level of the CLC legend was considered. Table 1 shows how this reclassification was carried out.

**Table 1.** Reclassification of CLC level II classes into BC classes.

| CLC—Level I | CLC—Level II | BC Classes |
|---|---|---|
| 1. Artificial surfaces | 1.1 Urban fabric | Built-up land |
| | 1.2 Industrial, commercial and transport units | Built-up land |
| | 1.3 Mine, dump, and construction sites | Built-up land |
| | 1.4 Artificial, non-agricultural vegetated areas | Built-up land |
| 2. Agricultural areas | 2.1 Arable land | Crop land |
| | 2.2 Permanent crops | Crop land |
| | 2.3 Pastures | Grazing land |
| | 2.4 Heterogeneous agricultural areas | Crop land |
| 3. Forest and seminatural areas | 3.1 Forest | Forest land |
| | 3.2 Shrub and/or herbaceous vegetation | Grazing land |
| | 3.3 Open spaces with little or no vegetation | Grazing land |
| 4. Wetlands | 4.1 Inland wetlands | Water (inland) |
| | 4.2 Coastal wetlands | Not included |
| 5. Water bodies | 5.1 Inland waters | Water (inland) |
| | 5.2 Marine waters | Not included |

Source: Our elaboration on CLC (2018), Global Footprint Network (2021).

Each area was converted into a bioproductive surface, measured in global hectares, through the equivalence factor ($EQF_i$) and yield factor ($Yw_i$) coefficients for Italy in 2016 [4] (see Table 2). A global hectare (gha), which is the accounting unit for the EF and BC, is a hectare with the world average biological productivity for a given year [4]. Equivalent factors convert one of the five land types into a standard unit of biologically productive area, represented by one gha. A yield factor accounts for the level of productivity of a given land type in a specific country with respect to the average world productivity.

**Table 2.** Values of equivalence factor and yield factor.

| Land-Use Type | Equivalence Factor (gha/ha) | Yield Factor |
|---|---|---|
| Built-up land | 2.522 | 0.767 |
| Crop land | 2.522 | 0.767 |
| Grazing land | 0.457 | 1.908 |
| Forest land | 1.286 | 1.679 |
| Water (inland) | 0.368 | 0.897 |

Source: Global Footprint Network, 2020.

Then, the per capita bio-productive area $BC_j$ of the municipality j is calculated by dividing the biocapacity of each municipality by its population, using the following equation:

$$BC_j = \frac{\sum_{i=1}^{5}(S_{ij} \times Yw_i \times EQF_j)}{Pop_j} \tag{3}$$

The resulting per capita values of $EF_j$ and $BC_j$ were compared to assess the $EB_j$ per capita for each municipality. Municipalities with $EB_j < 0$ were considered unsustainable, while those with $EB_j > 0$ were marked as sustainable. The results of the sustainability assessment are graphically illustrated by means of a set of maps, wherein municipalities with different values of EF, BC and EB are marked with different colors to highlight the areas of the country where a strong sustainability condition is achieved or is lacking.

## 4. Results

Figures 1 and 2, respectively, show the spatial distribution of $EF_j$ and $BC_j$ among Italian municipalities.

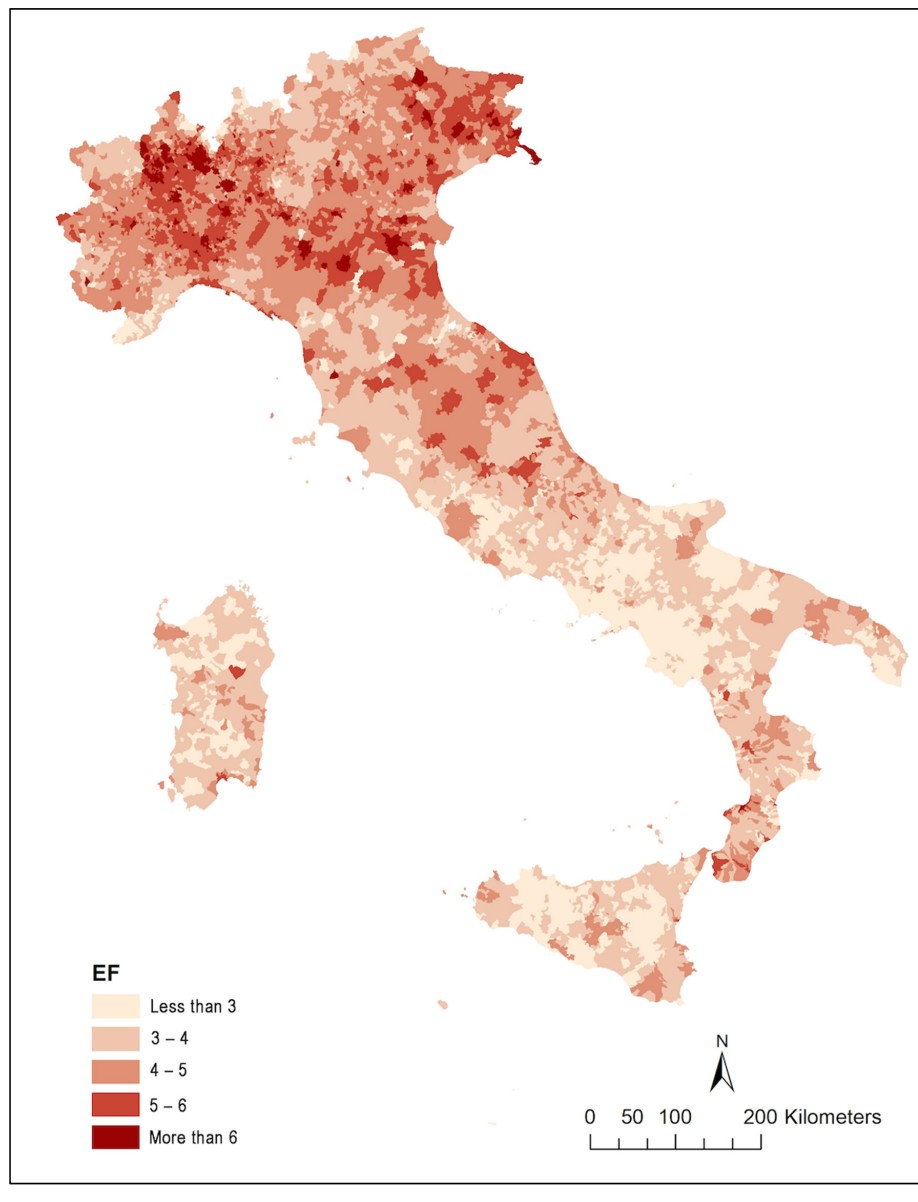

**Figure 1.** Distribution of per capita ecological footprint ($EF_j$).

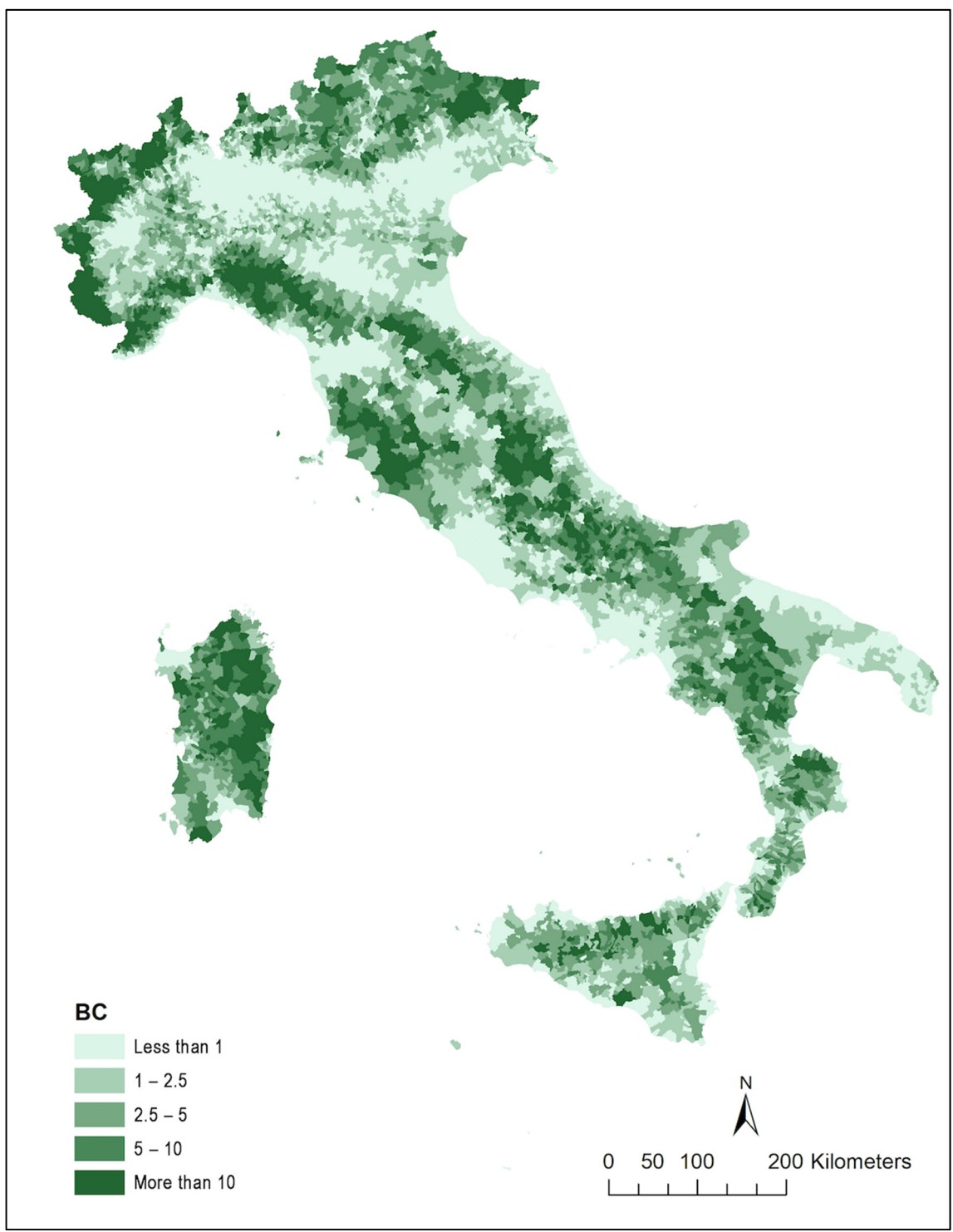

**Figure 2.** Distribution of per capita biocapacity (BC$_j$).

The EF$_j$ distribution (Figure 1) reports in spatial terms the (potential) level of consumption of the local residents evaluated with respect to the national situation. In fact, following the posited assumptions, the combination of per capita income and price index is the weight factor used to estimate the average individual ecological footprints of municipalities' residents. To derive a better interpretation of this map, it should be considered that the Italian per capita EF in 2017 is 4.41 gha [4]. This suggests that the municipalities, which are in the range of 4–5 gha, more or less have an ecological footprint in line with the national one. Differently, in the first two classes, with an EFj less than 4 gha, are included

those with a lower level of consumption, while the opposite situation characterizes those municipalities wherein the EFj is higher than 5 gha. The fact that, in general terms, the estimated levels of consumption are higher in northern Italy, despite the price levels being lower in the southern regions, confirms that the greatest effect on results is to be attributed to the average income of citizens.

The $BC_j$ distribution (Figure 2) is directly related to the municipalities' population density weighted by the relative prevalence of the different land-use categories. Consequently, the higher values of biocapacity are located in correspondence with the main mountain ranges (Alps and Apennines) and in the large rural areas of central and southern Italy. In contrast, low levels of biocapacity characterize the urban areas, in particular the ones located around the main Italian cities (Rome, Naples and Milan), the northeast industrial district and the flat regions of Emilia–Romagna, Tuscany and Puglia. In addition, it should be considered that the average Italian biocapacity in 2017 was 0.88 gha per capita [4], and, consequently, only the municipalities falling in the first class have a lower value of BC. This implies that, even if less than one-third of Italian municipalities have a below-average level of biocapacity, in some areas the availability of natural resources is so scarce as to influence the national figure.

Figure 3, which represents the main outcome of the study, shows the distribution of the ecological balance indicator, and can be interpreted as the map of Italian sustainability. The municipalities colored in shades of orange-red are unsustainable ($EF_j > BC_j$), and the ones in shades of yellow-green are sustainable ($EF_j < BC_j$). Looking at the map, it can be observed that the distribution of sustainable/unsustainable municipalities broadly follows the $BC_j$ distribution; the high level of association between BC and EB is confirmed by a correlation coefficient of 0.891. This is a consequence of the different scales of variation in EF (which ranges approximately from 2 to 8 gha) and BC (which ranges from 0 to more than 100); this gives BC greater influence in determining the final EB value.

Figure 4 shows the frequency distribution of Italian municipalities with respect to their EB value. The distribution is very asymmetric, and a long tail on the right is observed, related to the municipalities with very high values of BC. Consequently, the mean (−0.18 gha) and the median (−2.04 gha) of the $EB_j$ distribution provide two quite different indications of the general tendency of Italian municipalities' sustainability.

Further inferences of the outcomes of this study can be derived from the synthetic figures reported in Table 3.

**Table 3.** Data on sustainable/unsustainable municipalities, area, and population in Italy.

| Condition | Municipalities | | Area | | Population | |
|---|---|---|---|---|---|---|
| | Number | % | km$^2$ | % | Million | % |
| Sustainable | 2314 | 28.6% | 118,654 | 39.3% | 3.159 | 5.2% |
| Unsustainable | 5778 | 71.4% | 183,419 | 60.7% | 57.638 | 94.8% |
| Total | 8092 | 100.0% | 302,073 | 100.0% | 60.797 | 100.0% |

The negative value of the median implies that more than half the Italian municipalities are not sustainable; nevertheless, the fact that the share of municipalities with $EB_j < 0$ is 71.4% is quite impressive. As regards the spatial dimension, more than 39% of the Italian territories are sustainable municipalities, showing a positive value of EB. As far as the Italian population is concerned, the vast majority (almost 95%) live in municipalities characterized by unsustainable conditions.

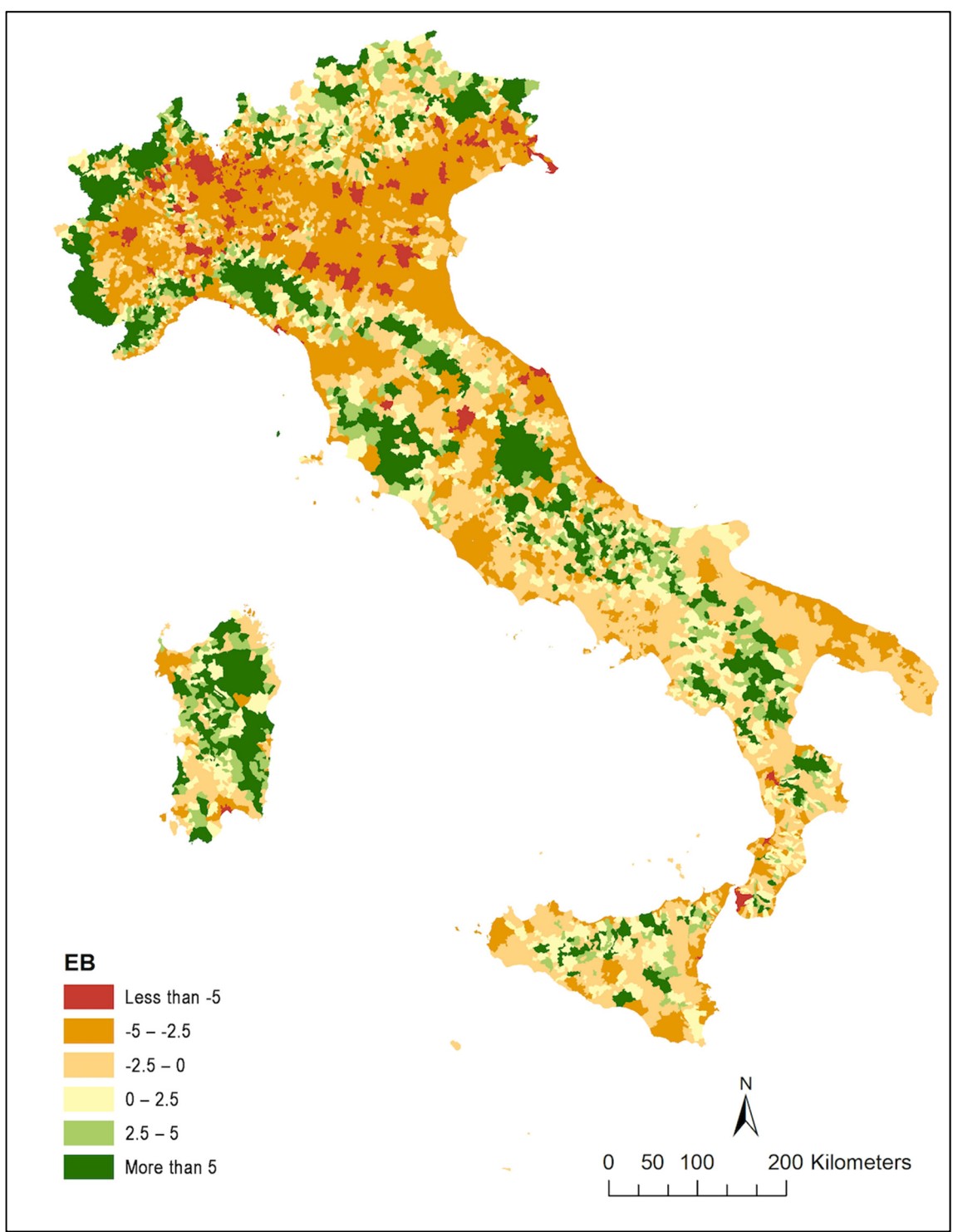

**Figure 3.** Distribution of ecological balance—EB$_j$ (gha).

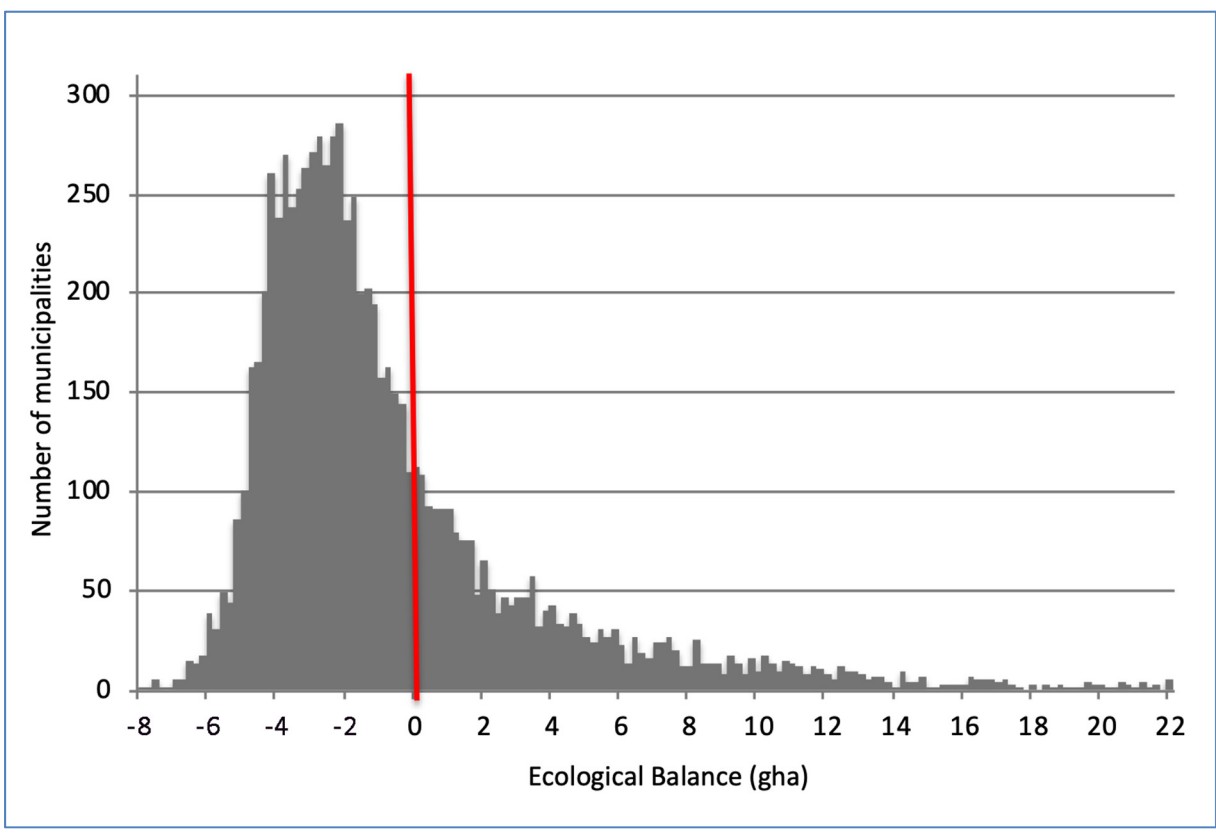

**Figure 4.** Frequency distribution of the municipalities by per capita EB (gha).

### 5. Discussion

These results are closely linked to the ecological footprint approach and to the methodology applied in this study to evaluate the EB. In this analysis, the role of the spatial scale is crucial, as it greatly influences the correct interpretation of the sustainability condition. Focusing on a small scale, such as the municipality, offers more detailed information about the territory.

The ecological balance of Italy, as a whole country, has a value of −3.53 gha per capita [4]. This study shows that this synthetic figure hides a wide set of different situations, with possible implications for better addressing environmental policies. To this end, two aspects should be considered. The first point is represented by the fact that almost 95% of the Italian population live in unsustainable municipalities, a figure that evidences the widespread local anthropogenic pressure on natural resources. The second point is linked to the possibility of identifying areas that produce positive externalities, thus partially compensating for the unsustainability of other territories.

The distribution of sustainable municipalities is more scattered, especially in northern Italy, where the role of the Alps and the less industrialized areas is evident in increasing the sustainability of the municipalities nearby. The sustainable areas grow in size as one travels south along the Apennines. A "green" spot emerges in the heart of Tuscany, in a region called Maremma, characterized by a historic vocation of extensive agriculture. The situation in Calabria, Sicily, and Sardinia is quite peculiar, as the inner part of each region is mostly sustainable, while the municipalities along the coast are not. This is linked to the fact that the inland areas of these regions are characterized by the presence of mountainous reliefs and vast agricultural areas. This determines, on the one hand, the concentration of the population along the coasts (flatter and better served by communication routes) and, on the other hand, the greater biocapacity of inland areas linked to land uses (forest, crop, grazing land).

From this point of view, ecological footprint accountability provides useful information in planning environmental policies in different stages of the decision-making process [49]. This was reported also by [50], who adapted the policy cycle of [51], highlighting the usefulness of the ecological footprint in each phase of the policy-making process. Indeed, EF can be very useful in the early warning phase, allowing the identification of ecological hot spots that need to be addressed. In our study, the identification of the most unstainable areas could drive national environmental policies to more targeted interventions.

Similarly, in the monitoring phase, where there is a need to assess the evolution of the environmental problems and the eventual effect of the adopted policies, EF can make a positive contribution, highlighting the possible effects of the implemented policies.

Furthermore, EF—given the immediacy with which it gives results—can also be useful in the headline and issue-framing phase, where it might be necessary to make a comparison among regions and to raise stakeholder awareness (in our case mainly at the local level).

A lower significance can be assigned to the policy development phase, linking the specific environmental policies with the general strategic policy framework, while for the implementation phase, EF appears to not be useful at all.

Figure 5 summarizes the relation between the steps of the policy cycle and the usefulness of information provided by the EF accountability approach.

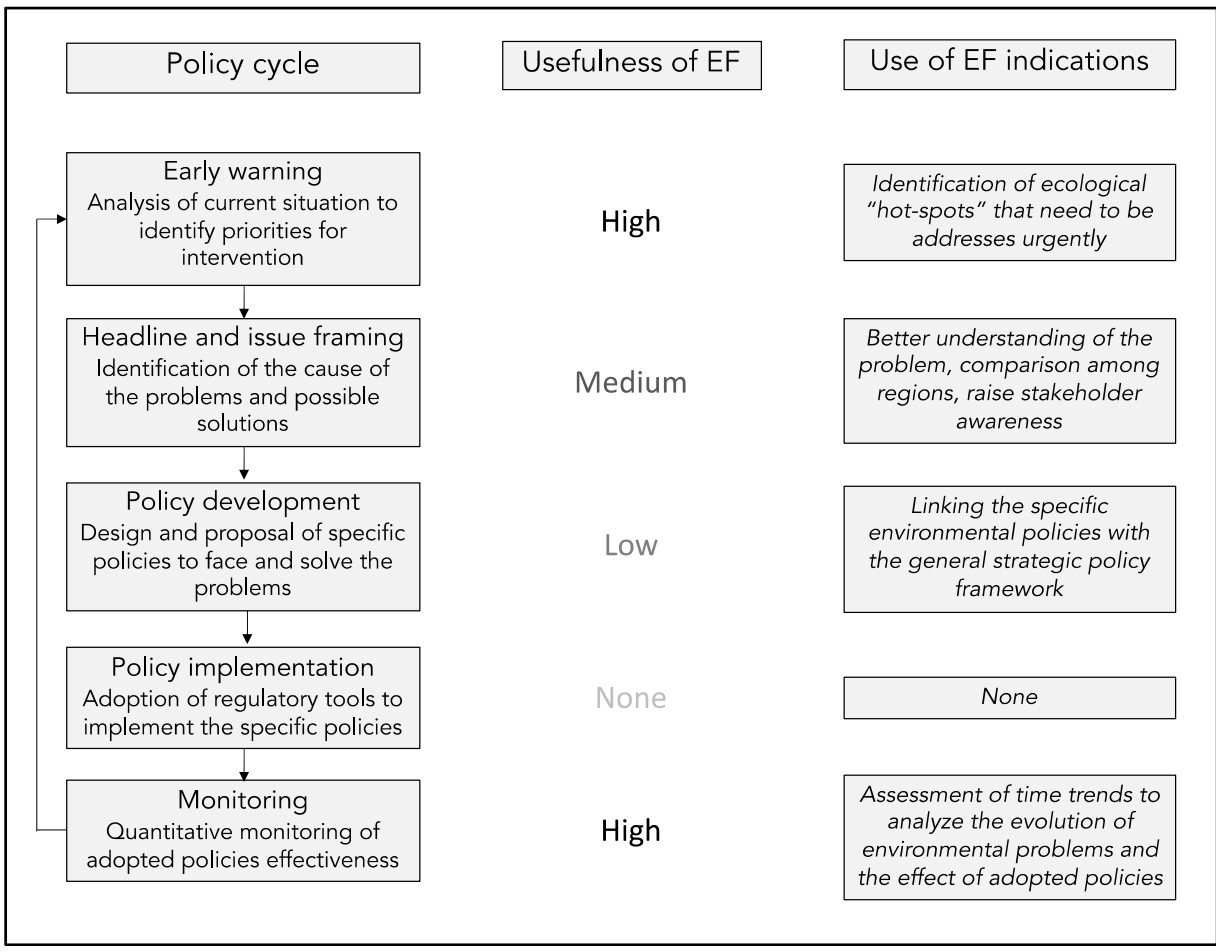

**Figure 5.** Policy usefulness of the ecological footprint based on the scheme proposed by [50].

## 6. Limitations

The outcomes of the study may be criticized for a few reasons.

The first point concerns the methodological approach; in fact, as pointed out in the paper, the ecological footprint, even if it is recognized by many scholars as a reliable indi-

cator for sustainability evaluation, has some conceptual limitations and, especially when applied on a local scale, requires quantitative simplification; for instance, the interconnectivity between municipalities, and how footprints within a municipality might be (more) influenced by adjacent populations and municipalities, particularly if resources are more heavily consumed elsewhere.

The second aspect is linked to the hypothesis underlying the local EF calculation, which is based on the assumption that the average purchase power of a municipality population directly relates to its level of consumption, which, in turn, determines the EF value. As pointed out in the methodology, this assumption, even if it is quite reasonable and supported by some studies, introduces a simplification into the local ecological balance calculation, and then into the Italian map of sustainability. It is evident, indeed, that EF not only depends on the quantity of consumption, but also on the typologies of purchased goods, and the income and price levels do not exactly reflect the amount of consumption.

Another limitation of the study that should be highlighted is the way in which BC is calculated. Indeed, the assessment of BC per capita in each municipality was based on the land types' bioproductivity (yield factors), assessed at the national level. In other words, the assessment of municipal ecological balance does not consider the variability in the local bioproductivity of cropland, forest land and grazing land. Such a limitation could be overcome in future studies by calculating the average productivity of the different land types on a regional scale.

## 7. Conclusions

This study aimed at mapping sustainability, using the ecological footprint approach as a tool to design a framework for environmental policy planning. Its outcomes are to be considered from a technical perspective. Despite the presence of numerous international policies that pay attention to environmental aspects, including the New Green Deal and the Sustainable Development Goals, there is a dearth of operational instruments that can help policy-makers in carrying out their function.

Despite the intrinsic limits of both the descriptive capacity and the calculation method of the ecological footprint, as raised by many authors, we consider that this indicator of sustainability has the advantage of being applicable at any scale, and it can show in quantitative terms the ecological balance of a territory. Consequently, it can provide a useful framework to identify specific areas of intervention, as it clearly shows where the highest anthropogenic pressure on an ecosystem occurs.

Going beyond considerations of the generalized unsustainability of the Italian territory and, consequently, the urgent need to promote specific environmental policies at the national level, the study highlighted substantial differences in the spatial distribution of the demand and supply of natural resources. In the applied methodology, the demand for natural resources is linked to national consumption styles, weighted by the local situation in terms of residents' real purchase power, while the supply is defined by municipalities' bio-productivity, divided by their population.

Environmental sustainability has potentially great implications for human welfare, and hence it represents a key goal of local policies. In the definition of these policies, indicators are becoming an increasingly essential tool. Their use is no longer limited to monitoring the progress of policies' implementation; rather, they assume crucial importance in the policy planning and decision phases.

This study, while providing preliminary results and presenting important limitations, emphasizes that the environmental sustainability of anthropogenic activities at the local level is affected by three main drivers: population density, residents' lifestyles and the bioproductivity of different land uses. This suggests that, if policy-makers actually intend to pursue the goal of environmental sustainability, their interventions should cover different areas; among these, as suggested by this study, priority should be assigned to policies aimed at redefining residential models, consumption behaviors and, last but not least, land-use patterns.

**Author Contributions:** Conceptualization, S.F. and B.P.; Data curation, A.M.; Methodology, S.F. and A.M.; Writing—original draft, S.F.; Writing—review & editing, B.P. All authors have read and agreed to the published version of the manuscript.

**Funding:** This research received no external funding.

**Institutional Review Board Statement:** The research is compliant with the Ethical Code of the University of Viterbo "La Tuscia".

**Informed Consent Statement:** Not applicable.

**Data Availability Statement:** Not applicable.

**Conflicts of Interest:** The authors declare no conflict of interest.

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
