# Peer review of "Mapping National Environmental Sustainability Distribution by Ecological Footprint: The Case of Italy"

_sustainability, doi:10.3390/su13158671_

Round 1

Reviewer 1 Report

This paper is well constructed and provides a moderately interesting mapping of ‘ecological footprint’, ‘biocapacity’ and ‘ecological balance’ by municipality for Italy. The paper would benefit from some editing by a native English speaker to tidy up the language.

I am unconvinced of the authors’ claims that this is a useful tool for policy-makers. Territorial self-sufficiency with respect to biocapacity is not normally a policy priority for local governments. Similarly, the authors’ argument that sustainability should be defined in such a way that it is “an attribute that can characterize a specific object as sustainable” fails to acknowledge that sustainability or unsustainability are characteristics of whole systems, and that objects within a system cannot be independently assessed as sustainable.

This objection does not remove all the value from the study. As the authors argue, ‘ecological footprint’ has been used widely to deem countries to be consuming less or more than their biocapacity, and therefore “sustainable” or not. In as much as this paper adds to this literature, it provides a useful disaggregation to depict the range of values present in one country.

In my view, it says little about municipalities’ sustainability, nor about useful policy measures to move toward sustainability. Indeed, it can mislead, such as the comment that “the need to provide targeted policies to improve the sustainability conditions clearly emerges almost 95% of the population lives in unsustainable municipalities, a condition of anthropic pressure mainly caused from a highly unbalanced population distribution.” If this is interpreted to mean that a more equal distribution of people over Italy’s territory would improve its sustainability, I beg to differ. Such a move would probably increase the average ecological footprint per capita.

One problem with Ecological Footprint methodology is that it includes fossil fuel use as if it were a renewable resource, substitutable for local biocapacity. (Another problem is that it omits other non-renewable resources, and pollutants other than carbon dioxide.) For any developed country, the majority of the footprint per person consists of fossil fuel use, which is entirely extraterritorially sourced in most jurisdictions. While a negative EB has implications for climate change (or would if BC did not grossly overestimate the capacity of terrestrial photosynthesis to draw down fossil fuel emissions by omitting to factor in the respiration of natural ecosystems), it says nothing about the overuse of natural resources locally. Hence I am personally not enamoured of the use of this methodology at local or even national scale. Global Footprint has been exceedingly effective in raising public awareness of human overshoot, but is less appropriate for rigorous scientific treatment of sustainability, which must disaggregate ecosystem services and identify limiting factors.

The paper relies on the calculations done by the Global Footprint Network of average Italian ecological footprint per capita. This is varied by the ‘potential consumption’ based on per capita income and a price index. Similarly, (but less precisely explained in the paper, so I’m unclear of the details) the biocapacity utilises national estimates, distributed by broad land-use classes with the help of a geospatial database of land use. As far as I can see, there is no attempt to quantify differences in biocapacity per hectare of cropland or grazing land in different parts of the country. Thus, the Ecological Footprint per capita map (Figure 1) is an effective map of socioeconomic inequality, while the Biocapacity per capita map (Figure 2) is reflective of little more than population density. The most policy-relevant question, of how to reduce ecological footprint of lifestyles at any given level of per capita purchasing power, is not examined because purchasing power is used as a surrogate for ecological footprint.

It should be noted that the definition of BC (“biocapacity”, given on p 4, second paragraph) does not state that it is per person, and its usage in Equation 1 does not refer to per person. It doesn’t need to, since the denominator would be the same for all terms, so the equation is equally valid whether it is per municipality or per person. But the annotation “BCj” suggests that it is for the whole municipality (J). Hence it caused me some confusion that the rest of the paper uses the term “biocapacity” to mean “biocapacity per person”. This is obvious only by reference to the caption for Figure 2. Where the word “biocapacity” refers to biocapacity per person, they should always state that it is.   

Some aspects of the methodology also need more explanation. The reader should not have to consult a reference to understand the derivation of ‘yield factor’ and ‘equivalence factor’. The terms “gha” and “wha” (Table 2) are not defined.

Regardless of my low regard for the usefulness of Global Footprint Network’s methodology applied at a local level, I think the paper should be published if the language is tidied, methodological descriptions are completed, and claims of policy relevance are either substantiated with demonstrably useful examples or removed.

Reviewer 2 Report

See the pdf file.

Round 2

Reviewer 2 Report

Overall, the study improved somehow after the revision.

The authors clarified the minor points.

As for the two major points:

(1) This was clarified, though the justification does not seem to me much stronger.

(2) This has improved slightly. I still do not think that the authors provided a robust argument how the study can contribute to "designing targeted environmental policies" (p. 13). They did not provide examples of such policies. Also, I do not think that the results can show "widespread anthropogenic pressure on local ecosystems" (p. 13) as most of the ecological footprint/deficit is caused by carbon emissions that do not directly affect local resources. 

Options for point (2): One possibility is to eliminate the link to policies altogether. The other possibility is to bring stronger arguments (which is likely to be difficult). Finally, another option is to keep the current version of the manuscript (the link to policies is less strong than in the first version of the manuscript). I leave this decision to the editor.
